# Assessing the Policy gaps for achieving China's climate targets in the Paris Agreement

Kelly Sims Gallagher [1], Fang Zhang[1], Robbie Orvis[2], Jeffrey Rissman [2] & Qiang Liu[3]

China committed to peak its carbon emissions around 2030, with best efforts to peak early, and also to achieve 20% non-fossil energy as a proportion of primary energy supply by 2030. These commitments were included in China's nationally-determined contribution to the 2015 Paris Agreement on climate change. We develop and apply a mixed-method methodology for analyzing the likelihood of current Chinese policies reducing greenhouse gas emissions in accordance with China's Paris commitments. We find that China is likely to peak its emissions well in advance of 2030 and achieve its non-fossil target conditional on full and effective implementation of all current policies, successful conclusion of power-sector reform, and full implementation of a national emissions-trading system (ETS) for the power and additional major industrial sectors after 2020. Several policy gaps are identified and discussed.

[1] Climate Policy Lab, The Fletcher School of Law and Diplomacy, Tufts University, 160 Packard Ave., Medford, MA 02155, USA. [2] Energy Innovation, 8 Battery St #202, San Francisco, CA 94111, USA. [3] National Center for Climate Change Strategy and International Cooperation, GuoHong Mansion, No. 11A, Muxidibeili, XiCheng District, Beijing 10038, China. Correspondence and requests for materials should be addressed to F.Z. (email: Fang.Zhang@tufts.edu)

This paper examines whether or not existing Chinese climate change policies are sufficient to enable China to peak its emissions around 2030 and to increase the share of non-fossil fuels in primary energy consumption to 20% by 2030. These are two of the targets that were contained in China's nationally-determined contribution (NDC), which was submitted to the United Nations Framework Convention on Climate Change (UNFCCC) secretariat under the Paris Agreement in June 2015[1]. To the extent that existing and forthcoming climate change policies are not sufficient to achieve China's NDC under the Paris Agreement, we aim to clarify the policy gaps that would have to be addressed for China to honor its commitments. We identify two types of policy gaps. The first is the discrepancy between current climate policies and the combination of current and additional policies that would be required to achieve China's NDC targets. The second is the disparity between how policies were designed and how they are implemented.

In the negotiation process leading to the Paris Agreement, all countries were requested to submit intended nationally-determined contributions (INDCs) to the UNFCCC. The United States and China announced their INDCs together in a joint statement by President Xi Jinping and President Barack Obama in November 2014[2]. These INDCs were converted into NDCs after the Paris Agreement entered into force in November 2016. Many governments, including China's, lacked clarity about the extent to which their country's existing and forthcoming policies would lead to emissions reductions domestically when they needed to determine their INDCs, leading to the submission of relatively conservative INDCs.

Past studies on China's ability to peak its $CO_2$ emissions have concluded that with China's current policies (not including new additional policies), China would either be able to peak before 2025[3], peak between 2025 to 2030[4], peak around 2030[5], or peak after 2030[6-10] based on methods including the Kaya identity analysis[3,5], the FAIR/TIMER model[6], computable general equilibrium models (CGE), bottom-up models[6,9], and an integrated assessment model (IAM)[10].

Declining $CO_2$ emissions in 2014 in China led some researchers to postulate that the peak may have already been reached. Some claimed that the decline of Chinese emissions is structural and is likely to be sustained if the growing industrial and energy system transitions continue[11]. Others believed that 2014 may just have been a short-term corrective dip resulting from macro-economic conditions and action to address conventional air pollution. Newly-published data on China's $CO_2$ emissions has indeed revealed a rise in China's $CO_2$ emissions in 2017, as predicted by some previous researchers[12], confirming that it is still too early to determine whether China's $CO_2$ emissions have peaked. The fluctuations of China's $CO_2$ emissions in recent years and the discrepancy between actual emissions and the results of previous modeling approaches demonstrates the need for new methodological approaches to predict China's emission trajectory with specific policies implemented.

The existing body of research identifies various factors driving China's peaking path, including economic growth, industrial structure, energy intensity, the energy mix, technological change, and the population growth rate. The power system reform is important as the fundamental pre-condition for China to peak its $CO_2$ emissions[13]. Continuous effort in strengthening energy efficiency will prove to be the game changer, underpinning the impacts of China's 2030 Energy Revolution Strategy[14]. China's new development model is the key driver of China's downward trajectory of greenhouse gas (GHG) emissions[3]. Industrial structure and the decarbonization of the energy system serve as the two most important drivers of emissions reductions, followed by decreasing energy and emission intensities[11]. Rational urbanization should be another key factor[15]. Finally, readers should note that frequent modifications to energy statistics in the past have led to considerable uncertainty about China's ability to achieve its carbon mitigation targets[16]. The data used in this paper were validated by the official Chinese government think tank, the National Center for Climate Change Strategy and International Cooperation, ensuring that they are the latest statistics available.

With this research, we endeavor to clarify more precisely which existing climate policies have been most effective at limiting emissions in China, and how they interact with each other in doing so. We present a mixed-method methodology for assessing the policy gaps that we define at the beginning of the paper. We provide a replicable methodology for any country trying to determine its climate policy gaps. Both expert elicitation and a system dynamics model are employed to explore the first type of policy gap where the absence of specific policies leads to a discrepancy between the current policy package and the policy package that is required by a country to achieve its NDC targets. The expert elicitation also works to identify the second type of policy gap, where the poor design or implementation of existing policies in practice leads to the inability of a country to achieve its NDC targets. The system dynamics modeling tool has already been developed for other countries and is designed to be easily adapted. An extension of the mixed method utilized in this paper would be to analyze the likely impact of prospective new policies as countries begin to formulate their future targets and mid-century strategies.

We find that China is likely to peak its emissions well in advance of 2030 and achieve its non-fossil target based on current policies, conditional on full and effective implementation of all current policies, successful conclusion of power-sector reform, and full implementation of a national emissions-trading system (ETS) for the power and additional major industrial sectors after 2020. Implications of these findings are that the Chinese government should focus on fully implementing existing policies, completing the power sector reform as soon as possible, implementing and strengthening the national ETS, making energy efficiency standards more stringent in the future, and developing new carbon pricing policies for non-covered sectors.

## Results

**Policy inventory**. To determine which policies to include in the analysis we developed a comprehensive policy inventory. We classified climate policies as explicit or implicit. Explicit policies are those that would not be implemented for any other reason other than to reduce GHG, such as a cap-and-trade program for $CO_2$ emissions or a regulatory performance standard for $CO_2$ emissions. Implicit policies are those that were implemented for multiple reasons not exclusive to addressing climate change, but that have the effect of reducing GHG emissions (e.g., China's feed-in tariffs for renewable energy or afforestation policies). We further categorize climate policies by type[17]. The main types are regulatory/administrative, fiscal, market-based, informative, innovation, diplomatic, and other. Some policies fit into more than one of these categories. The policy inventory yielded more than 100 separate climate policies at the national level in China, which we categorized by type (Supplementary Data 1) and have made available in Supplementary information [https://figshare.com/s/3cc9d39b26155714b0eb].

**Expert elicitation**. To determine which policies were likely the most important in reducing greenhouse gas emissions to date and in the future (and which were not) we conducted a qualitative survey of Chinese and foreign climate policy experts. The

qualitative survey was conducted during June/July 2017, and in October 2018. Two-thirds of the respondents were Chinese experts and one-third were foreign experts. All of the experts believed that China would peak in advance of 2030, but three-quarters of them believed it would be difficult or relatively difficult to peak well in advance of 2030. The non-fossil fuel target was perceived to be more challenging from the experts' perspective and three-quarters of them thought it would be difficult or impossible for China to achieve the 20% non-fossil fuel target. Regarding the existence of a climate policy gap, all experts believed that a policy gap currently exists, but that the gap is not large and can be fixed with modest efforts to reform existing policies or to implement new ones.

These experts identified 17 policies that they believed were the most influential in limiting $CO_2$ emissions to date. The six policies that were most frequently cited were: feed-in tariffs for renewable energy, energy efficiency standards for power plants and motor vehicles, non-fossil energy targets, mandated caps on coal consumption, energy efficiency standards for buildings and equipment, and the key enterprise program for energy efficiency. The full list of influential emission-reducing policies generated by the survey is provided in Fig. 1.

A number of policies were also identified by individual experts as either no longer necessary or in need of reform as summarized in Fig. 2: the national emission trading system (ETS), power sector reform (incomplete), electric vehicle subsidies (too high), resource tax (too low and not covering $CO_2$), feed-in tariff (replace with ETS or carbon tax), manufacturer subsidies, and CCS promotion policies. There was no consensus among experts about which policies were no longer necessary or in need of revision.

Experts were also asked which new policies were needed most to achieve the NDC targets, and the three most frequently

- Subsidies for renewable energy (feed-in tariffs)
- Energy efficiency policies for power plants
- Non-fossil targets
- Coal cap policy
- Energy efficiency standards for buildings and equipment
- Key enterprise program
- Economic structural reform
- Removing fossil fuel subsidies
- Energy intensity targets
- Green bonds
- Air pollution standards
- Differentiated electricity tariffs
- Phasing out old and inefficient power plants
- Electricity sector reform
- Innovation policies
- Environmental resource tax
- National emissions-trading system (ETS)

**Fig. 1** Most important climate policies through 2016. As we gave experts the freedom to list the climate policies that they thought were important through 2016 (rather than sticking to the list selected policies provided in the appendix), some experts introduced new policies that are not included in the appendix and the system dynamics model, including the coal cap policy, energy intensity targets, green bonds, air pollution standards, and phasing out old and inefficient power plants

mentioned responses were: policies to reduce non-$CO_2$ gases, a carbon tax for sectors not covered under the new national ETS, and entrepreneurship incentives for low-carbon firms as indicated in Fig. 3. China's NDC only covers $CO_2$, however, so the first policy recommendation would not, in fact, help China to achieve its NDC even though it would limit overall GHG emissions. Experts expressed mixed views about China's emerging ETS, with some being skeptical that it would produce a $CO_2$ price that would incentivize behavioral change, and others feeling more confident about its eventual success over time. Few experts identified innovation policies, economic reform policies, or industrial policies as influential in emissions reductions, but the authors of this paper believe that these policies are, in fact, key to the ability of China to limit future emissions. Innovation policies, although indirect and difficult to model, are likely to become even more important in the future. Innovation policies include investments in energy research, development, and demonstration; investments in human capital; and creation of market-pull incentives for creating new products and processes that will contribute to GHG emission reductions.

**System dynamics model.** The policies identified by experts as being most influential in limiting Chinese $CO_2$ emissions were included in a system dynamics model developed collaboratively between U.S.-based Energy Innovation and China's National Center for Climate Change Strategy and International Cooperation (NCSC). Two additional modifications to existing policy were also introduced into the model because they have already been announced but not implemented: power sector reform based on least cost dispatch (switching from the current guaranteed dispatch policy to a policy of marginal cost-based dispatch in the electricity sector) and elimination of mandated electricity capacity construction targets (switching from a government planning to a market mechanism for adding new electricity capacity). Power sector reform is underway in China but it remains incomplete. Mandated electricity capacity construction targets for coal-based power capacity must be eliminated to solve the problem of overcapacity in the electricity sector. While eliminating capacity targets is formally envisioned in Chinese power sector reform policy documents, it has not been consistently adhered to in practice. In addition, the model incorporates technological innovation as an endogenous variable and simulates the effects of a research and development (R&D) policy through a technological learning mechanism.

To capture the impacts of all these current and envisioned policies on China's total $CO_2$ emissions, we constructed two scenarios. One is a counterfactual business-as-usual (reference case), where no climate change policies are employed. The other one is a climate policy package scenario, where all 14 existing and forthcoming climate change policies are included. The reference case is largely based on outputs from other models. The policy package scenario was developed by the Climate Policy Lab at Tufts University based on the policy inventory and expert elicitation. The assumptions included the reference case and the policy package scenario are described in the methods section.

The model is designed to avoid double-counting the effects of individual policies in order to accurately assess the interactive effect of the total policy package. In this study, policies that may be undertaken to reduce emissions are either price-driven or are governed by a separate policy lever particular to that action, but not both. There are essentially two ways to avoid double counting: either the separate policy lever is specifically defined to be additive to any price-induced shifting, or the separate lever is a price floor (or ceiling) that only takes effect after price-induced shifting occurs. For instance, the model adjusts the electric vehicle

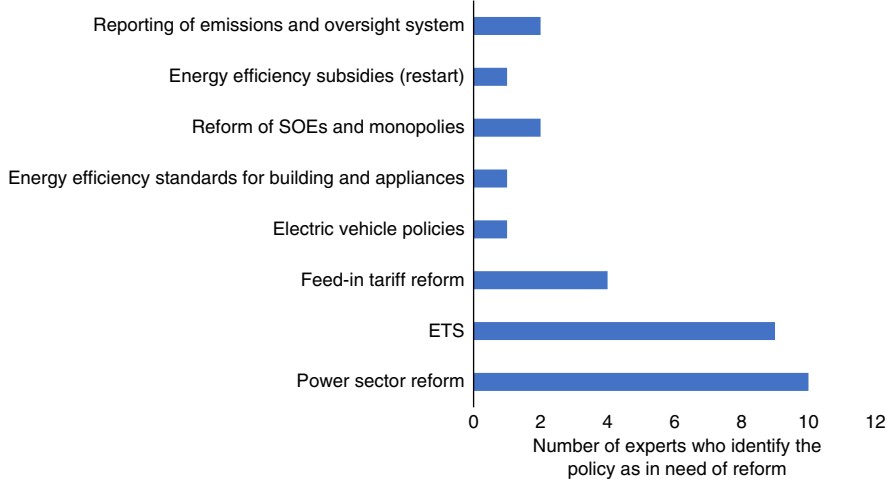

**Fig. 2** Policies that are most in need of reform. SOE refers to state-owned enterprises. ETS refers to the national emission trading system for the power sector. As we gave experts the freedom to list the climate policies that they thought were most in need of reform, some experts introduced new policies that are not included in the appendix and the system dynamics model. The results show that power sector reform, ETS and the feed-in-tariff for renewables are the top three policies that are most in need of reform

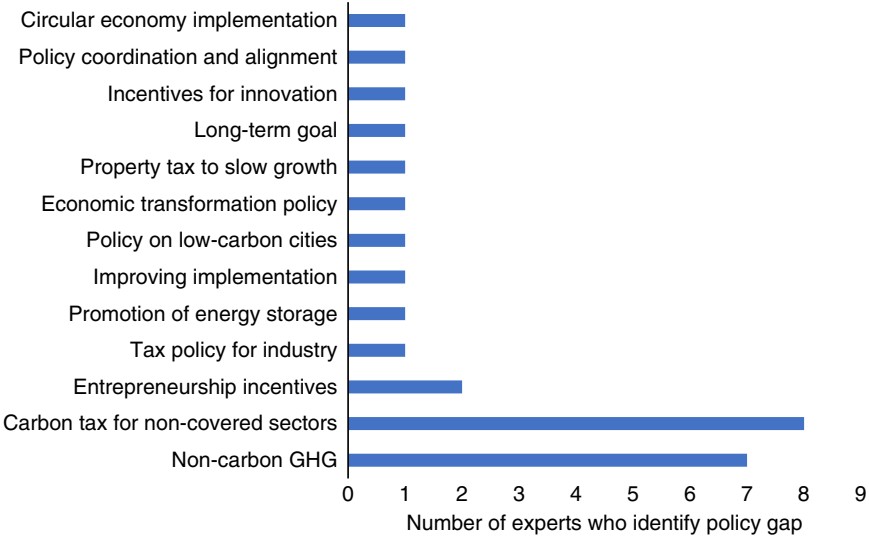

**Fig. 3** New policies needed to close the climate policy gap in China. GHG refers to greenhouse gas. Carbon tax for non-covered sectors and non-carbon GHG (e.g., CH4) are recognized as the two most important new policies needed. Notably, there is little consensus on the magnitude of other kinds of new policies that are needed to close the climate policy gap in China

(EV) market share based on policies that affect fuel price, reflecting the way fuel prices would influence buyers' vehicle choices. The model also includes an EV sales mandate, which can require that at least a certain percentage of vehicle sales consist of EVs. The EV sales mandate is implemented as a floor, so the mandate has no effect if EV sales would be high enough to comply based on pricing policies alone.

The results of the policy package scenario show that China's $CO_2$ emissions from energy combustion, industrial processes, and land use, land use change, and forestry (LULUCF) peak twice, which is different from previous predictions (Fig. 4). The first peak is a temporary, small peak in 2019 at level of 10.7 billion tonnes. There is a subsequent permanent peak in 2026 at the level of 11.8 billion tonnes, well ahead of the 2030 target. After 2026, a long-lasting plateau ensues until 2040 rather than a decisive post-

peak drop in emissions. The first peak occurs as the result of the implementation of power-sector reforms that require marginal-cost dispatch. Initially, in this scenario, there is surplus capacity in electricity supply. The second peak occurs when the overcapacity problem is finally resolved and the new electricity supply is derived from non-fossil sources.

The results of the policy package scenario indicate that existing climate change policies will limit growth in China's absolute energy consumption but will not enable it to reach a peaking point (Fig. 5). Total primary energy use in China continues to increase but gradually transitions to a slower pace after 2040. To achieve the peak in energy use envisioned by the IEA and Reinventing Fire scenarios, new and additional policy instruments would be required. Regarding the structure of energy use, the non-fossil target of 20% is almost achieved at 19.86% by 2030

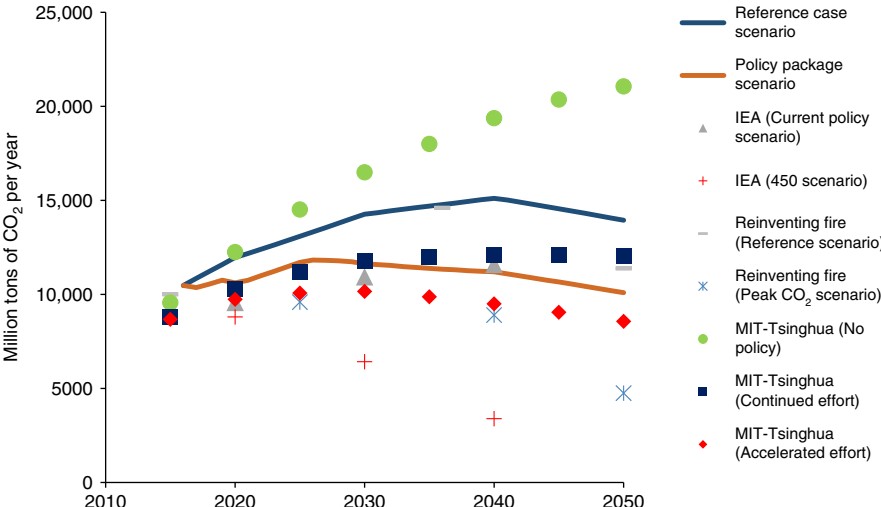

**Fig. 4** Total $CO_2$ emissions in China under different scenarios. The results of the policy package scenario show China's $CO_2$ emissions from energy combustion, industrial processes, and land use, land use change, and forestry (LULUCF) peak twice, which is different from previous predictions. The first peak is a temporary, small peak in 2019 at the level of 10.7 billion tons. There is a subsequent permanent peak in 2026 at the level of 11.8 billion tons, well ahead of the 2030 target. After 2026, a long-lasting plateau ensues until 2040 rather than a decisive post-peak drop in emissions. Meanwhile, the $CO_2$ emission comparisons between the scenarios of our model and the scenarios of the other models[8,9,18] indicate the robustness of our model setting. The $CO_2$ emission level of our reference case scenario is between that of MIT-Tsinghua (No policy) and Reinventing Fire (Reference scenario). The $CO_2$ emission level of our policy package scenario is higher than the levels of MIT-Tsinghua (Continued effort) and IEA (Current policy) as we include more climate change policies, but higher than the levels of the Peak $CO_2$ scenario of Reinventing Fire, MIT-Tsinghua (Accelerated effort) and IEA (450 scenario) where new policies beyond current policy package are added

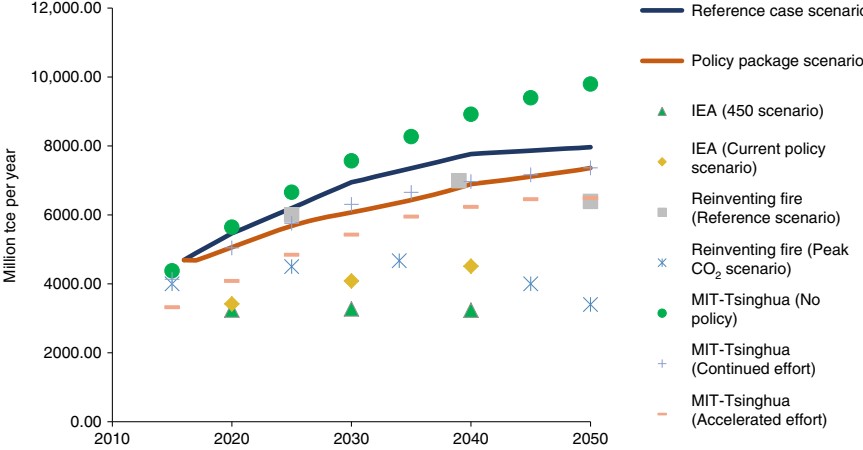

**Fig. 5** Total primary energy use in China under different scenarios. The results of the policy package scenario in our modeling show that existing climate change policies will limit growth in China's absolute energy consumption but will not enable it to reach a peaking point. Total primary energy use in China continues to increase but gradually transitions to a slower pace after 2040. To achieve the peak in energy use advanced by the IEA[18] and Reinventing Fire scenarios, nine more stringent policies would be needed

(Fig. 6). China is likely, therefore, to meet its commitment of 20% of primary energy consumption from non-fossil fuels by 2030 if all current policies are fully implemented as planned.

We find that no single policy results in the achievement of China's $CO_2$ emissions peak by 2026 (Fig. 7). The combination of policies most influencing emissions in 2030 are power sector reform, industrial transformation, industrial efficiency, ETS, and light-duty vehicle efficiency. Though our modeling draws attention to the importance of the power sector reform, this policy will not be the biggest factor before 2030 (as depicted in Fig. 7). The dispatch mechanism from guaranteed dispatch to a least cost dispatch in the power sector reform should facilitate the installation and use of renewables. The transition from an

administrative approach to a market-based system will also solve the substantial overcapacity in China's power sector.

**Comparing the expert elicitation and model results**. Both the expert elicitation and system dynamics modeling methods produce results that indicate China is likely to peak in advance of 2030. The modeling results are more optimistic as they show that China could peak well in advance of 2030, whereas a majority of the experts interviewed believed that it would be difficult or not easy for China to peak well in advance of the target date. Both methods produce results that imply that it will be challenging for China to fully achieve the 20% non-fossil fuel target without additional effort (although the modeling results indicate no

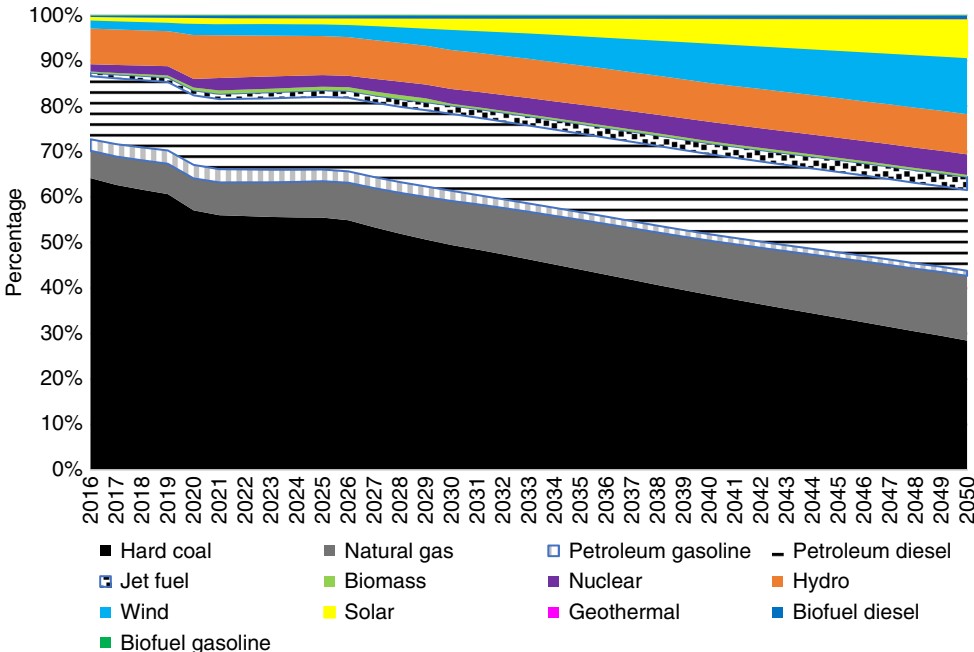

**Fig. 6** Primary energy use by fuel type in the policy package scenario. Bright colors show the share of primary energy use in China from non-fossil fuels, including biomass (light green), nuclear (dark purple), hydro (orange), wind (blue), solar (yellow), geothermal (purple), biofuel diesel (dark blue), and biofuel gasoline (dark green). The total share of non-fossil fuels in China's primary energy consumption gradually increases and will reach to 19.86% by 2030. China will, therefore, nearly meet its commitment of 20% of primary energy consumption from non-fossil fuels by 2030

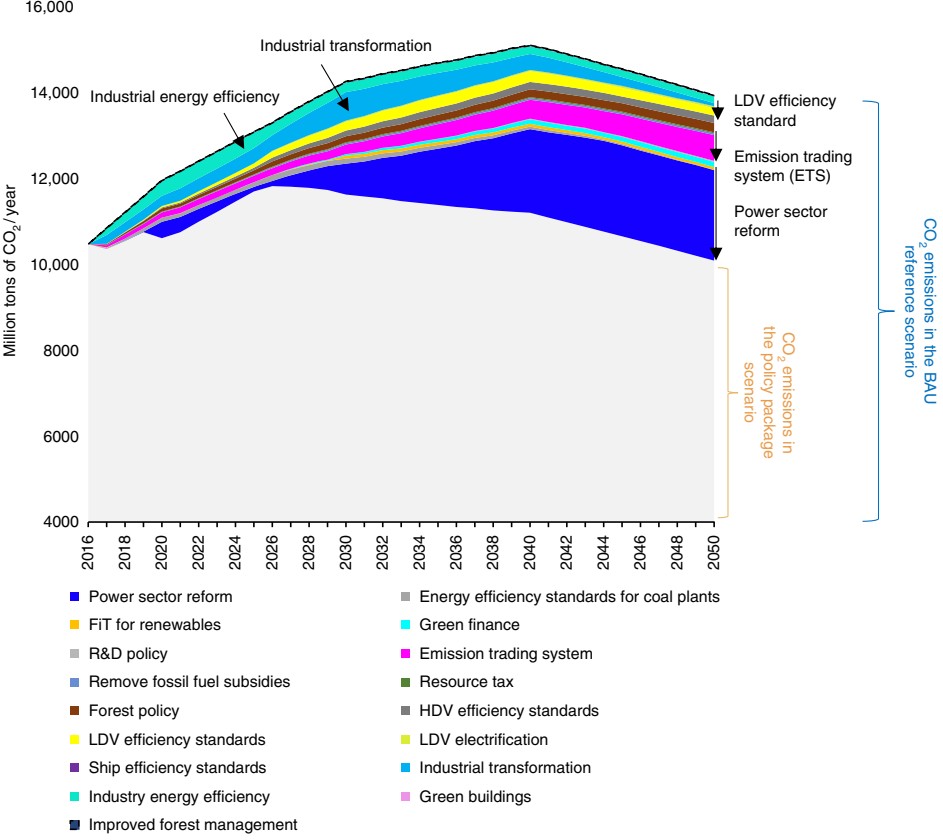

**Fig. 7** Annual $CO_2$ reductions by policy in the policy package scenario relative to the BAU reference case scenario. R&D refers to research and development. LDV refers to light-duty vehicle and HDV refers to heavy-duty vehicle. The various colorful segments show how much each policy can reduce $CO_2$ emission. The results show that no single policy results in the achievement of China's $CO_2$ emissions peak by 2026. The top five most effective policies before 2030 are power sector reform, industrial transformation, industrial efficiency, ETS, and LDV efficiency

meaningful gap to achievement of the target because full implementation of the policies is assumed in the modeling). The expert elicitation results indicate that several policy reforms are needed, particularly for the feed-in tariffs, national emissions trading system, and power sector reform process.

The expert survey and modeling results both indicate that power sector reform, industrial transformation, industrial efficiency, the ETS, light-duty vehicle (LDV)/heavy-duty vehicle (HDV) efficiency, afforestation, and energy efficiency standards for power plants are all essential for the achievement of China's climate targets by 2030. The expert elicitation also emphasizes the need to focus on non-$CO_2$ GHG emissions, removing fossil fuel subsidies, expansion of the green bond finance policy, innovation policy, LDV electrification, and green building standards. Afforestation is more important in the model than experts believe it to be. Both sets of results agree that China's current small resource tax currently has a negligible effect on $CO_2$ emissions.

## Discussion

Many previous studies suggest that the largest potential for $CO_2$ emissions reductions depend upon the decrease in energy consumption per unit GDP derived from improvements in energy efficiency and changes in economic structure[3,11,14]. In this study, improvements in energy efficiency aren't a single intervention, but are implemented through individual policies across different sectors. If we add them up, their cumulative impacts are substantial. In addition, we assess the impacts of current efficiency standards, but not any new ones even though it is likely that Chinese policymakers will continue to update the standards in the future. The impact of economic structural change in this study appears to be smaller than others[3,11] have estimated. The reason for this discrepancy is that we only evaluate the impact of specific climate policies, not the impact of underlying economic structural change. Our BAU reference case does assume a modest amount of natural economic structural change that would happen without the policy package. For instance, we assume an economic growth rate of 6.5% between 2016 and 2020, which then gradually decreases to 2.5% by 2041–2050. The industrial sector's contribution to GDP is assumed to decrease from 35% in 2020 to 31% by 2050.

Power sector reform reveals itself to be a precondition for decarbonizing the Chinese electricity sector so that non-fossil electricity sources can be fully deployed on the grid. Traditionally, the electricity pricing schemes in China were administratively determined and they favored incumbents over resource and cost-neutrality, which put coal plants in a more favorable position. To reflect the ongoing but gradual process of power sector reform in China, the model assumes that the power sector reform will slowly phase-in beginning in 2020 and be fully implemented by 2027. In that same year, electricity pricing switches to a market-based mechanism, which allows for deeper electricity decarbonization. After being introduced in 2017 in the model, the national ETS initially has a very modest impact on $CO_2$ emissions reductions due to the modest price signal modeled ($7/ton $CO_2$, increasing 3% annually through 2030). But as the model shows, the role of the ETS could become significant and influential over the longer term if it can sustain a much higher emissions trading price. The effectiveness of the ETS also depends on the full implementation of power sector reform, as do other fiscal policies like the feed-in tariffs. Post 2025, if the ETS cap is tightened, this carbon pricing instrument could become a major driving factor for emissions reductions in the power sector. Energy efficiency standards, particularly for coal-fired power plants and industry, but also for motor vehicles, will remain very important during the

next ten years. Efficiency standards will need continuous updating to continue driving progress on reducing carbon intensity.

Methodologically, the mixed-methods approach increases confidence in the results, which should inform policymaker decision-making about which policies to reform and which new policies should be developed and implemented. There are, however, several uncertainties and limitations which should be highlighted for accurate interpretation of the study results. First, the impacts of policies are very sensitive to the assumptions made about the specific policies used to achieve the emissions reductions and how those policies are implemented. Clearly, assumptions about the possible range of policy instruments and how well they might be implemented can lead to large differences in the resulting reductions in $CO_2$ emissions. Second, the uncertainty regarding the actual effects of individual policies effects is likely the smallest when a fewer number of policies are used. Uncertainty increases as a greater number of policies are introduced. Because we included 14 policies in the model, the degree of uncertainty was thus increased compared with a smaller hypothetical policy package. In addition, as in all system dynamics modeling, the model we employ for this study relies on relationships among variables. Any bias around a given variable or coefficient can be multiplied and lead to system uncertainty due to the multiple feedback mechanisms. Another risk is that we can only include a limited number of variables to realistically simulate the effect of policy on China's emissions. Our modeling includes more than 2000 variables, but we are still unable to model certain types of policies (e.g., entrepreneurship incentives for low-carbon firms and reforms of state-owned enterprises) or design details within certain categories of policies. For instance, the current design of China's ETS is effectively a tradable performance standard. Our model treats the ETS in China as a carbon price, which does not precisely capture the actual design features of the ETS in China. We believe our study might also underestimate the impacts of economic reform, innovation policy, and building efficiency improvements.

In conclusion, we find that an early peak in $CO_2$ emissions seems likely, but there is little room for complacency since we assume full implementation of existing and announced policies. Similarly, if power sector reform is completed and mandatory capacity construction targets for coal-based power capacity are eliminated, we find that China practically achieves its non-fossil targets. We note that energy efficiency policies are crucial in enabling China to peak its emissions earlier and will need to be continuously upgraded over time. Power sector reform is also very important to achieve the non-fossil target, achieve deeper reductions in $CO_2$ emissions over time, and to enable other other policies to work well. The new national ETS for the power sector, while important, is not currently sufficient to induce major emission reductions because of the anticipated low prices and narrow sectoral coverage in the near term. Economic reform, innovation policies and energy efficiency policies for buildings are likely to be more important than are currently revealed in the model. Finally, it is important to not be overly optimistic in modeling the future BAU reference-case scenario.

## Methods

**A mixed method approach.** A mixed method approach was employed combining qualitative and quantitative approaches to understanding China's policy gaps for achieving its Paris Agreement targets. Three primary methods were used. The first was to develop an inventory of all national-level policies that were promulgated after 2000, categorized by type and issuing ministry. The second was an expert elicitation of both Chinese and foreign climate change policy experts who were administered a survey with 18 questions about the effectiveness of the policies identified in our policy inventory. The third was to develop a system dynamics model specific to China where individual policies identified in our policy inventory are introduced and are allowed to interact with one another. Notably, the expert

**Table 1 Background information on experts surveyed**

| Expert | Institutions | Expertise |
|---|---|---|
| 1 | Tsinghua University | Energy and climate change modeling, energy, and climate change policy |
| 2 | Tsinghua University | Climate change, energy, and climate change policy |
| 3 | Tsinghua University | Climate change, energy, and sustainable development |
| 4 | Renmin University | Energy and environment modeling, energy, and climate change policy, international climate institutions |
| 5 | Chinese Academy of Science | Sustainable development, climate change, energy package, environment governance and policy |
| 6 | Chinese Academy of Social Science | International climate governance, energy, and climate policy |
| 7 | Energy Foundation, China | Climate change modeling, climate change policy, technology innovation and transfer |
| 8 | Energy Foundation, China | Climate change, clean energy, and clean transportation |
| 9 | National Center for Climate Change Strategy and International Cooperation | Energy and renewable energy policy and strategy |
| 10 | National Center for Climate Change Strategy and International Cooperation | Climate change strategy and policy, low carbon development |
| 11 | Development Research Center of the State Council | Industrial economics, climate change |
| 12 | Former Ministry of Environment | Climate change governance |
| 13 | Former Ministry of Finance | Green finance and climate change |
| 14 | Massachusetts Institute of Technology (MIT) | Air quality & health, energy, climate policy, regional analysis |
| 15 | Syracuse University | Environment economics, economics of technological change |
| 16 | Lawrence Berkeley National Laboratory | Low-carbon development modeling, building energy efficiencies, appliance efficiency standards |
| 17 | Lawrence Berkeley National Laboratory | Energy and environment modeling, energy, and climate change policy, international climate institutions |
| 18 | Georgetown University | Environmental policy, technology transfer |

**Table 2 Policy instrument comparison between the BAU reference case and the policy package scenario**

| Policy instruments | BAU reference | Policy package scenario |
|---|---|---|
| R&D policy | Existing level R&D expenditure | Doubling R&D expenditure for clean energy sector by 2020 |
| Emission trading system (ETS) | 0 | Start with $7/ton in power sector and chemicals in 2017 and includes aluminum and others after 2020. Prices grow by 3% annually |
| LDV efficiency standards | 7l/100KM by 2020;6l/100 KM by 2050 | 5l/100KM by 2020; 4l/100 KM by 2025 |
| HDV efficiency standards | n/a | Close to the global advanced level by 2020 (the energy efficiency increases by 15% by 2020) |
| Ship efficiency standards | n/a | The energy efficiency increases by 20% by 2020 from 2005 levels |
| LDV electrification | n/a | EV takes up 20% percent of car sales in 2025 |
| Resource tax | 6% | 7.5% of price for coal and natural gas. Petroleum and oil not included |
| FiT for renewables | 0 | Wind: 0.127 RMB/KWh; Solar: 0.334RMB/Kwh; Biomass: 0.189RMB/KWh; Geothermal: 0.327RMB/Kwh. All linearly phasing out by 2030 |
| Industrial transformation | n/a | Speeds up the service sector's ratio by another 2 percent before 2030 |
| Industry energy efficiency | n/a | Energy efficiency targets for heavy industrial sectors in the 13th Five Year Plan (FYP) for iron and steel, cement, chemical, paper etc |
| Energy efficiency standards for coal plants | n/a | 300 g standard coal equivalent per KWh for new power plant by 2020 |
| Power sector reform | n/a | Linearly phase in least cost dispatch from 2020–2027 and all least cost dispatch after 2027; 2017–2020 phasing out 5 GW coal capacity; overcapacity is solved by coal sector from 2020–2030; after 2030, market-based |
| Green finance policy | n/a | Interest rates reduced by 2% for renewables (Wind, PV, thermal, hydro, nuclear) by 2030 (impacts on energy efficiency not included) |
| Forest policy | n/a | Forest coverage increased to 23.4% by 2020 and 26% by 2030; Restoration of degraded forest by 10 million hectare during 2016–2020; restoration of degraded forest by 48.750 hectare during 2021–2050 |

R&D policy only includes wind, PV, thermal, hydro, nuclear, not bio-mas and does not include energy efficiency. RMB refers to Renminbi, which is Chinese currency

elicitation process was separate from the modeling process. The modeling proved most helpful to clarify which policies could lead to a discrepancy between the current set of policies and the required policy package that is required for China to achieve its NDC targets. The expert elicitation proved helpful in identifying the second type of policy gap, where policies that were promulgated were not well implemented, pointing to the need for reform or enhanced implementation of those policies.

**Policy inventory**. The policy inventory is attached in Supplementary Data 1 and also available in online Supplementary information via figshare.com

[https://figshare.com/s/3cc9d39b26155714b0eb]. The Climate Policy Lab at the Fletcher School of Law and Diplomacy, Tufts University, assembled the inventory mainly from primary sources, namely government documents issued by each relevant government ministry in China. Secondary sources, such as the International Energy Agency's Policies & Measures database, were also used.

**Expert elicitation.** For the expert elicitation, thirty-six top experts on China's energy and climate policies were asked to complete a written survey and a follow-up interview was offered to all experts. They were selected because of their capacity to understand the big picture, and their ability to answer our questions regarding which policies are relatively more important than others. We did not target experts who are famous for any niche issue within China's climate change policy, such as issue experts on transportation, building efficiency or afforestation. Eighteen responses were received, thirteen of which were from Chinese experts, and five from foreign experts. Background information about these experts, including institutions and expertise are provided in Table 1. The survey was semi-structured and we provided experts with the freedom to identify climate policies that they determined were important through 2016, rather than requiring them to stick to the predetermined list of policies provided in the Appendix attached to the survey protocol. Some experts identified policies that were not included in our policy inventory. We uploaded the survey protocol (Supplementary Note 1) as on-line Supplementary information via figshare.com [https://figshare.com/s/483d43b02d6110c8ea24], so the survey is now available for review. The study protocol was approved by the Institutional Review Board (IRB) at Tufts University since the expert elicitation method involved human participants. We obtained consent from all participants during the fieldwork.

Most Chinese experts preferred to respond to the questions orally, and their responses were recorded by hand. Gallagher and Zhang conducted these interviews in either Chinese or English (or a mixture of both). All the foreign experts chose to respond with written responses to the survey and declined the follow-up interview.

**Modeling.** There are three main reasons why we chose to use a system dynamics model for this study. First, this simulation technique stresses the feedback dynamics of stocks and flows and the associated time delays in achieving objectives, which enables the capture of the interactions among policies. Second, the system dynamics model is capable of evaluating a wider scope of policy instruments—including both pricing and non-pricing policy instruments as it focuses on disequilibrium dynamics and feedback complexity, rather than on equilibrium and optimal factor allocations. Third, there is currently no other research that employs a system dynamics model to evaluate the likelihood of China meeting its NDC targets under its current policy framework.

In addition, system dynamics models have been increasingly employed to conduct national energy policy evaluation, energy efficiency analysis, to assess the development of the energy industry, and to predict carbon emissions trajectories[19–22].

Most Chinese climate models are either CGE models such as IAM and China-in-Global Energy Model (C-GEM), or bottom-up models such as The Integrated MARKAL-EFOM System (TIMES) and Long-range Energy Alternatives Planning System (LEAP). None of the existing models fully reflect all of the major climate policies that, in reality, are already influencing carbon emissions in China. CGE models reflect climate policies that work through pricing mechanisms under the assumption of a perfectly efficient economy and often, for simplicity, use a carbon price as a proxy for all types of climate policy[3,8]. Bottom-up models can include more detailed, sector-level climate change policies but often fail to capture macro-level climate change policies and the interactions among policies.

Energy Innovation and the NCSC constructed the initial version of the system dynamics model. The name of this model is the China Energy Policy Simulator. China-specific data for the model was mostly collected from public sources or supplied by NCSC. A public web-based version of the model is available at http://china.energypolicy.solutions. The model structure is completely open source and it has been reviewed by other institutions, including Argonne National Lab and Lawrence Berkeley National Lab. Readers can experiment with the online model to get a sense of how this model operates. Some assumptions to the BAU reference scenario were updated from the version created by Energy Innovation and NCSC for the purpose of this study, including reduced business as usual capacity for renewables, updated capacity factors for different generation technologies, and adjusted business-as-usual fuel economy standards for passenger vehicles. All policies were coded using assumptions about the annual stringency and time frame of each policy based on current literature. All policies can be turned on individually or collectively with others so that the interactions among them can be taken into account. To assess whether China's climate policies can achieve its targets, we only modeled two scenarios: one is the reference case scenario, which is business-as-usual where no climate change policy is employed, and the other one is the policy package scenario, where all existing and forthcoming climate change policies are turned on (Table 2). The policy package scenario was developed by the Climate Policy Lab at Tufts University based on the policy inventory and expert elicitation.

**Ethical compliance statement.** The study protocol was approved by the Institutional Review Board (IRB) at Tufts University since the expert elicitation method

involved human participants. We obtained consent from all participants during the research fieldwork.

## Code availability

All code is available upon reasonable request.

## Data availability

All of the raw data from the expert elicitation cannot be available due to Institutional Review Board (IRB) regulations protecting human subjects. A public web-based version of the system dynamics model is available at http://china.energypolicy.solutions. Authors are willing to entertain requests for underlying data in the model.

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

## Acknowledgements

This work was carried out with the support of Energy Foundation China, The William and Flora Hewlett Foundation, and BP, Inc. Research assistance for the policy inventory was provided by Qi Qi, a graduate student at The Fletcher School.

## Author contributions

Kelly Sims Gallagher developed the overall research question, contributed to the policy inventory, expert elicitation, and the system dynamics model. Fang Zhang contributed to expert elicitation and the system dynamics model. Robbie Orvis contributed to the system dynamics model. Jeffrey Rissman and Liu Qiang contributed to the initial version of the system dynamics model.

## Additional information

**Competing interests:** There is potential concern of non-financial interests since the fifth author is affiliated with the official Chinese government think tank on climate change. But this concern should not be overestimated as the research design was developed by an independent research institute and substantial modification to the initial version of the system dynamics model was made, to which the fifth author contributed. The Authors declare no further competing interests.

