## [Peer Review File · Nature Communications]

Reviewers' comments:

Reviewer #1 (Remarks to the Author):

Reviewer's comments to NCOMMS-18-22618-T

Summary of review: The paper investigated a very important policy issue in China's climate policy. The method shed some experiences for discussing similar questions in other countries context.

A few comments and questions for the paper:

1. Literature review: the current literature review is a bit casual and loose, and needs add the relevant paper citations. There are also conflicting reports on China's emission peak (already peaked vs. going to peak vs. continue growing), it would be helpful to position this paper by commenting on the authors view on the topic. Exemplified by the following papers:

- Guan, et al. 2018. "Structural Decline in China's CO2 Emissions through Transitions in Industry and Energy Systems." <https://doi.org/10.1038/s41561-018-0161-1>.
- Green and Stern. 2015. "China's 'New Normal': Structural Change, Better Growth, and Peak Emissions." http://www.lse.ac.uk/GranthamInstitute/wp-content/uploads/2015/06/Chinas_new_normal_green_stern_June_2015.pdf.
- Yuan, et al. 2014. "Peak Energy Consumption and CO2 Emissions in China." <https://doi.org/10.1016/j.enpol.2014.01.019>.

2. Expert solicitation: The paper shows the limit of sample size and other constraints, however, it would be useful to at least clarify how the experts are chosen, and if the experts agree, to discourse the institution, and their expertise. It is not clear what questions are asked. Is the survey structured, semi-structured, or open questions? The model input is some key assumptions. How are the survey results integrated into the model? With so many questions, it is not clear how the model would be different/more effective with or without the survey.

3. Model: The paper applies the system dynamic model to study the impact of the policy inventory, which is fine. However, it would be more persuasive to clarify why the authors think the system dynamic model is the best model to study this question. It may useful to include a comparison of the model and results with other peer models. For example, it is not clear how the RnD Renewable's contribution to emission reduction is calculated. How the model deals with overlaps of different policy instruments.

4. Uncertainty: system dynamic model heavily relies on the relations of variables which are highly uncertain. While the feedback is sometimes better taken, it may risk with system uncertainties. The paper needs to discuss those uncertainties in a more coherent and transparent fashion.

Overall, the paper tries to include the expert view into the system dynamic modeling which is an improvement of the modeling practice. I'd suggest improvement on the literature review, clarification on the expert interview, and uncertainties on modeling before it could be accepted.

Reviewer #2 (Remarks to the Author):

A Policy Gap Analysis for China's Climate Targets in the Paris Agreement

By Gallagher, Zhang, Orvis, Rissman, and Qiang

Submitted to NATURE, August, 2018

I found the objective of this paper to be interesting but struggled with the content. To begin with, the paper never really defines the paper's focal point, "a policy gap". The paper then is so short in length and in explanation that it seemed to me very ad hoc. Figures lack captions and labels on the axes, many acronyms are not defined, and figures 4 and 5 have so many elements in a range of colors that it is difficult to visualize the messages.

The paper calls on expert elicitation and a systems dynamics model to explore the effect of current and anticipated policies on China's CO2 emissions. Only in the on-line section on methods do we learn that the expert elicitation involved 11 responses from "top experts", with expertise that is not revealed. Inconsistencies in the categories of figures 2, 5, and 6 make it hard to determine exactly which policies were considered and which are currently in place. The ultimate question was explored with 2 scenarios, business-as-usual, and one in which all policies are fully implemented.

The conclusion that "China is on track..." is not very convincing when followed by the disclaimer that "this result assumes full and effective implementation of all current policies, successful conclusion of power-sector reform, and full implementation of a national emissions trading system (ETS)".

It seems to me that this manuscript is an interesting beginning, but not yet a mature paper ready for NATURE.

Reviewer #3 (Remarks to the Author):

I am delighted to review the manuscript, and appreciate the authors' efforts to identify the policy gap and map out the relative role of different policy measures for China to achieve its climate targets in the Paris Agreement. I would say that the research objective is very ambitious. It could answer the question to some extent, but might bring even more questions.

Currently I have following comments on the manuscripts.

- 1) "Our mixed methodology includes the use of expert elicitation and an innovative systems dynamics model, which overcomes many of the shortcomings of existing models of China." It is a very strong statement. I am convinced yet. For any policy assessment with a model like CGE and TIMES, it often involves substantial qualitative analytical work as well. CGE model and TIMES model have deficiencies, but they could to some extent characterize the decisions of producers and/or consumers and capture the mechanisms by which the policies affect their decisions, and ultimately the CO₂ emissions level. I do not think SD models have any advantages in this respect. Many years ago, a group in MIT Sloan School of Management developed a CO₂ calculator with SD model. But it is largely a calculator of CO₂ emissions, could not be regarded as a model for rigorous policy assessment.
- 2) According to Figure 5, power sector reform would be the largest contributor to the cross-economy CO₂ reductions. It is difficult to understand. Many studies suggest that the largest potential in CO₂ emissions lie with the decrease in energy consumption per unit GDP derived from improvements in energy efficiency and changes in economic structure. Power system reform can play an important in the deployment of renewable energy. Increased renewable uses, however, could not become the largest contributor to the cross-economy CO₂ reductions at least before 2030.
- 3) According to the current design of China's ETS, it would largely a tradable performance standard. It would have a substantial overlap with "Industry energy efficiency" and "Efficiency standards for coal plants". I don't know how authors could figure out the relative contribution of the three measures.

Response Letter

Reviewers' comments:

Reviewer #1 (Remarks to the Author):

Reviewer's comments to NCOMMS-18-22618-T

Summary of review: The paper investigated a very important policy issue in China's climate policy. The method shed some experiences for discussing similar questions in other countries context.

A few comments and questions for the paper:

1. Literature review: the current literature review is a bit casual and loose, and needs add the relevant paper citations. There are also conflicting reports on China's emission peak (already peaked vs. going to peak vs. continue growing), it would be helpful to position this paper by commenting on the authors view on the topic. Exemplified by the following papers:

- Guan, et al. 2018. "Structural Decline in China's CO2 Emissions through Transitions in Industry and Energy Systems." <https://doi.org/10.1038/s41561-018-0161-1>.
- Green and Stern. 2015. "China's 'New Normal': Structural Change, Better Growth, and Peak Emissions." http://www.lse.ac.uk/GranthamInstitute/wp-content/uploads/2015/06/Chinas_new_normal_green_stern_June_2015.pdf.
- Yuan, et al. 2014. "Peak Energy Consumption and CO2 Emissions in China." <https://doi.org/10.1016/j.enpol.2014.01.019>.

Thanks for this suggestion, as well as the relevant publication recommendations. The literature review in our previous manuscript had been highly abridged to meet the word limitation requirements of *Nature Climate Change*, which was the journal to which we originally submitted the paper before it was referred to *Nature Communications*. In the revised version, we have substantially expanded our literature review section. In response to this reviewer's suggestions, we specifically added discussion on the conflicting reports regarding China's emissions peak and on factors shaping China's peaking pathways.

2. Expert solicitation: The paper shows the limit of sample size and other constraints, however, it would be useful to at least clarify how the experts are chosen, and if the experts agree, to discourse the institution, and their expertise. It is not clear what questions are asked. Is the survey structured, semi-structured, or open questions? The model input is some key assumptions. How are the survey results integrated into the model? With so many questions, it is not clear how the model would be different/more effective with or without the survey.

We have provided more detail regarding the expert elicitation in the method section. We promised anonymity to the experts, but have added a new table (Table 1) that provides the institutional affiliation and substantive expertise for each surveyed expert. The experts are all well-known scholars with expertise regarding China's economy-wide climate change or energy policies. They were selected because of their capacity to understand the big picture, and their

ability to answer our questions regarding which policies are relatively more important than others. We did not target experts who are famous for any niche issue within China's climate change policy, such as issue experts on transportation, building efficiency or afforestation.

Though none of the reviewers explicitly requested it, we expanded the sample size and conducted additional expert elicitations. We conducted additional expert elicitations in order to test the robustness of the survey since our original sample size was relatively modest with 11 experts. Since August, we submitted the survey to an additional 5 foreign experts and 9 Chinese experts, and in total we received 7 responses. Again, the foreign experts filled out the survey and the Chinese experts preferred to be interviewed. So, we now have a total of 18 expert elicitations. The results from the expanded expert elicitation did not change significantly with the addition of more experts. The results only strengthen the analysis.

The survey was semi-structured and we provided experts with the freedom to identify climate policies that they determined were important through 2016, rather than sticking to the predetermined list of policies provided in the Appendix attached to the survey protocol. Some experts proposed new policies that were not included in our policy inventory. We uploaded the survey protocol as on-line supplementary information via figshare.com [<https://figshare.com/s/483d43b02d6110c8ea24>], so the survey is now available for review.

We clarified the relationship between the survey and modeling methods in the Methods section beginning on page 14. To be clear, the expert elicitation is a separate research method that we employed to clarify expert judgment about which climate policies have been most important to date in limiting or avoiding GHG emissions in China. The expert elicitation results also informed the modeling exercise because we ensured that all of the policies identified by experts were included in the system dynamic model.

3. Model: The paper applies the system dynamic model to study the impact of the policy inventory, which is fine. However, it would be more persuasive to clarify why the authors think the system dynamic model is the best model to study this question. It may useful to include a comparison of the model and results with other peer models. For example, it is not clear how the RnD Renewable's contribution to emission reduction is calculated. How the model deals with overlaps of different policy instruments.

We appreciate this recommendation, and elaborate on the shortage of existing modelling studies on Page 2 and explain why we chose to employ a system dynamics model to evaluate the impacts of climate change policy more in our text on Page 3, and also in the Methods section beginning on page 14 and ending on page 17. There are three main reasons why we chose to use the system dynamics model. First, this simulation technique stresses the feedback dynamics of stocks and flows and the associated time delays in achieving objectives, which enables the capture of the interactions among policies. Second, the system dynamics model is capable of evaluating a wider scope of policy instruments – including both pricing and non-pricing policy instruments as it focuses on disequilibrium dynamics and feedback complexity, rather than on equilibrium and optimal factor allocations. Third, there is currently no other

research that employs a system dynamics model to evaluate the likelihood of China meeting its NDC targets under its current policy framework.

We want to emphasize that we would not claim that the system dynamics model we use is superior to others. We also added total CO₂ emissions and total primary energy use to our previous two main figures (Figure 5 and Figure 6) to illustrate comparisons. We also added more discussion of our findings compared with those of others, especially regarding different views on policy instruments in the Discussion section beginning on page 12.

As suggested, we clarified how we calculate the renewables' R&D contribution to emissions in our research on page 7. Technological innovation is endogenous in this research. The cost reductions from endogenous learning are applied to the start year costs for wind and solar PV and other technologies. The effects of the R&D policy lever are considered through the mechanism of inducing technological learning, which results in a change in total construction cost per unit capacity, and in turn, shapes the deployment of these renewables.

As suggested, we also clarified how we are able to avoid double-counting policy effects on page 8, which is crucial to accurately assess the effects of a policy package consisting of many interacting policies. This study deploys essentially two methods to avoid double counting: either the separate policy lever is specifically defined to be additive to any price-induced shifting, or the separate lever is a floor (or ceiling) that only takes effect after price-induced shifting. For instance, the model adjusts EV market share based on policies that affect fuel prices, reflecting how fuel prices influence buyers' vehicle choices. The model also includes an EV sales mandate, which requires that a certain percentage of vehicle sales consist of EVs. The EV sales mandate is implemented as a floor, so the mandate has no effect if EV sales are high enough to comply based on pricing policies alone.

4. Uncertainty: All system dynamic models heavily rely on the relations of variables which are highly uncertain. While the feedback is sometimes better taken, it may risk with system uncertainties. The paper needs to discuss those uncertainties in a more coherent and transparent fashion.

It is true that system dynamics models are built upon the relationships of variables and that they must consider the feedback effects among variables. Therefore, the selection of variables and their coefficients are crucial. Any bias around a given coefficient can be multiplied. To minimize the uncertainty rising from selections of parameters, we referred to multiple peer-reviewed papers and other models when we set up the relationships among variables.

There are, however, uncertainties that could not be practically addressed. We added a section on uncertainties in the Discussion section to transparently clarify the limitations of the model, which should help readers interpret the results with more accuracy on page 13.

Overall, the paper tries to include the expert view into the system dynamic modeling which is an

improvement of the modeling practice. I'd suggest improvement on the literature review, clarification on the expert interview, and uncertainties on modeling before it could be accepted.

Thanks. Indeed, we are intentionally combining two methodological techniques in order to improve the robustness of the findings.

Reviewer #2 (Remarks to the Author):

A Policy Gap Analysis for China's Climate Targets in the Paris Agreement
By Gallagher, Zhang, Orvis, Rissman, and Qiang
Submitted to NATURE, August, 2018

I found the objective of this paper to be interesting but struggled with the content. To begin with, the paper never really defines the paper's focal point, "a policy gap". The paper then is so short in length and in explanation that it seemed to me very ad hoc. Figures lack captions and labels on the axes, many acronyms are not defined, and figures 4 and 5 have so many elements in a range of colors that it is difficult to visualize the messages.

Because we originally submitted to *Nature Climate Change* (before being referred to *Nature Communications*) we were subject to extremely tight word restrictions, which caused the paper to be highly truncated. In this revision, we clarified terms and acronyms (such as R&D, LDV, HDV, and EV), added a literature review, added much more discussion, and elaborated our findings more since we are allowed to double the word allowance, include more figures, and have an unrestricted number of words for the methods sections.

We added a definition of the "policy gaps" on page 1 when we first introduce this term. Specifically, policy gaps refer to the discrepancy between China's current set of climate policies and the policy package that is required by China to achieve its NDC targets. The discrepancy can result from either the absence of certain policy instruments or deficiencies in the design or implementation of specific policies which requires further policy reforms.

We modified the original Figure 4 and 5 (which are now numbered separately as Figure 7 and Figure 8). For Figure 7 on page 10, rather than depicting the cumulative amount of energy use, we provide percentages to show the changes in energy mix. We also used brighter colors for non-fossil fuels and darker colors for fossil fuels so that it will be easier for readers to digest the figures. We also added more explanatory text to Figure 8 to facilitate the understanding of this figure on page 11.

The paper calls on expert elicitation and a systems dynamics model to explore the effect of current and anticipated policies on China's CO₂ emissions. Only in the on-line section on methods do we learn that the expert elicitation involved 11 responses from "top experts", with expertise that is not revealed. Inconsistencies in the categories of figures 2, 5, and 6 make it hard to determine exactly which policies were considered and which are currently in place. The

ultimate question was explored with 2 scenarios, business-as-usual, and one in which all policies are fully implemented.

As already discussed in our response to Reviewer 1, we now clarify how experts were selected, and how the elicitation was conducted on page 14-15. With consent from the experts, we added a table to clarify the institutions and expertise of the individuals who completed the survey.

We added to the text after Figure 2 to clarify the apparent inconsistency between Figure 2 and Figure 8 (Figure 5 in the previous manuscript). In our revised manuscript, we explain that Figure 2 derives from the expert explication. As we used semi-structural interviews and gave experts the freedom to list the climate policies that they were though important through 2016 (rather than sticking to the list selected policies provided in the appendix), some experts introduced new policies that are not included in the system dynamics model, specifically, the non-fossil targets, coal cap policy, key enterprise program, air pollution standards.

The conclusion that “China is on track...” is not very convincing when followed by the disclaimer that “this result assumes full and effective implementation of all current policies, successful conclusion of power-sector reform, and full implementation of a national emissions trading system (ETS)”.

Thank you for encouraging us to be more precise in our conclusion. You are correct. We modified the wording of this claim. We find that China is likely to peak its emissions well in advance of 2030 and to achieve its non-fossil target based on current policies under the condition that all current policies are fully and effectively implemented, including successful conclusion of power-sector reform, and full implementation of the national emissions-trading system (ETS) for the power and additional major industrial sectors after 2020.

Reviewer #3 (Remarks to the Author):

I am delighted to review the manuscript, and appreciate the authors’ efforts to identify the policy gap and map out the relative role of different policy measures for China to achieve its climate targets in the Paris Agreement. I would say that the research objective is very ambitious. It could answer the question to some extent, but might bring even more questions. Currently I have following comments on the manuscripts.

1) *“Our mixed methodology includes the use of expert elicitation and an innovative systems dynamics model, which overcomes many of the shortcomings of existing models of China.” It is a very strong statement. I am convinced yet. For any policy assessment with a model like CGE and TIMES, it often involves substantial qualitative analytical work as well. CGE model and TIMES model have deficiencies, but they could to some extent characterize the decisions of producers and/or consumers and capture the mechanisms by which the policies affect their decisions, and ultimately the CO2 emissions level. I do not think SD models have any advantages in this respect. Many years ago, a group in MIT Sloan School of Management developed a CO2 calculator with*

SD model. But it is largely a calculator of CO₂ emissions, could not be regarded as a model for rigorous policy assessment.

As discussed in our response to the second question of Reviewer 1, we clarified the relationship between the expert elicitation and the system dynamic model in the new methods section on page 14-15. We freely acknowledge that the development of the CGE and TIMES models involves substantial and iterative qualitative analytical work. The inputs to all models (equations and parameter values) are based on “expert” opinion. The expert elicitation in this research is, however, was not primarily intended to help parameterize the model. The expert elicitation was intended to obtain a deep qualitative evaluation of policies, which provides an independent set of results. The qualitative approach allows us to compare the results from the expert elicitation with those from the modelling, and to identify both convergence and discrepancies. The expert elicitation is also used to identify the second type of policy gap (where a discrepancy between the intended policy design and policy implemented on the ground), which is impossible to capture in the model.

We agree that CGE model and TIMES model indeed could characterize the decisions of producers and/or consumers and capture the mechanisms by which policies affect their decisions, and ultimately the CO₂ emissions. But the way that the CGE model captures the impact of policy is through its effect on the relative prices of energy and other goods, which in turn affects fuel and technology choices, the composition of domestic economic activity, and global trade dynamics (Zhang et al., 2016). CGE models often assume that the policy is being introduced into a perfectly efficient economy in which there are no distortions or market failures. (Green and Stern, 2015). A further limitation of GCE models are that they do not capture the potential for climate policies to induce innovation in green technologies, with knowledge spillovers into other sectors, which are likely to drive higher GDP growth than otherwise (Aghion et al., 2014). CGE models are highly complex and have been thought as a black box (Pindyck, 2017). As a bottom-up approach, TIMES can include more detailed, sector-level climate change policies but are challenged in capturing macro-level climate change policies and the interactions among policies. Some of the models may also overestimate emissions reductions since multiple policies result in smaller reductions than they would if implemented individually.

The system dynamics has been increasingly applied to national energy policy evaluation, energy efficiency analysis, and the development of energy industry, and urban energy and carbon emission (Anand, Vrat, and Dahiya, 2006; Kunsch and Springael, 2008; Feng et al., 2013; Robalino-Lópezab, Mena-Nietob, andGarcía-Ramos, 2014). Systems dynamics models can have two major advantages to reflect the impacts of climate change policies. One is that this simulation technique stresses the feedback dynamics of stocks and flows and the associated time delays in achieving objectives and learning mechanisms, which enable the capture of interactions among policies. The other advantage is that the system dynamics model is capable of evaluating a wider scope of policy instruments – including both pricing and non-pricing policy instruments as it focuses on disequilibrium dynamics and feedback complexity, rather than on equilibrium and optimal factor allocations. To date, no other system dynamics model has yet

been employed to evaluate whether or not China's existing policies are likely to cause the achievement of China's NDC targets, which makes this effort a significant contribution. To better harvest the last benefit, we now compare the results of our model with those of others on page 9-11.

2) According to Figure 5, power sector reform would be the largest contributor to the cross-economy CO2 reductions. It is difficult to understand. Many studies suggest that the largest potential in CO2 emissions lie with the decrease in energy consumption per unit GDP derived from improvements in energy efficiency and changes in economic structure. Power system reform can play an important in the deployment of renewable energy. Increased renewable uses, however, could not become the largest contributor to the cross-economy CO2 reductions at least before 2030.

In our classification, improvements in energy efficiency are split up across different sectors. If we add them up, their impacts are substantial. In addition, we assess the impacts of CURRENT policies, which means that future efficiency standards are not assumed to be continuously updated. In fact, they are likely to be updated, and we provide a more detailed discussion of energy efficiency in the Discussion section. We agree economic structural change is very important in influencing emissions. The relative impact of economic structural change in our model is smaller than in other models because we only evaluate the additional economic structural change that is driven by specific policies. Our BAU reference case includes an underlying economic structure change that would occur even without the set of climate policies introduced into the model, and we added more specifics about the assumed growth rates and decline of the industrial sector to the paper.

We agree with Reviewer 3 that the power sector reform cannot have the largest influence before 2030, and our results actually produce this conclusion too (as shown in Figure 8). Our results indicate a much larger impact of power sector reform over the longer term. This relatively large impact of power sector reform does not only come from the deployment of renewables, but come from two other major factors. First, power sector reform enables more electricity generation from renewables and more efficient coal power plants by moving from a guaranteed dispatch system to a least cost dispatch system. The other factor is that the substantial overcapacity in China's power system is gradually resolved through the prevention of construction of new power plants. This occurs from moving from an administratively run electricity market to a market-based system. The former hinders the deployment of renewables in the short term. As a result, the impacts of the power sector reform are relatively weak before 2030. We added more discussion on Page 10-12 to explain these issues.

3) According to the current design of China's ETS, it would largely a tradable performance standard. It would have a substantial overlap with "Industry energy efficiency" and "Efficiency standards for coal plants". I don't know how authors could figure out the relative contribution of the three measures.

Yes. The current design of China's ETS would largely be a tradable performance standard, which will theoretically overlap with the existing industrial energy efficiency and coal power plant efficiency standards. Our model treats China's ETS as a traditional price-based ETS rather than performance standard-based one. There is therefore no double-counting between the ETS and industrial/power energy efficiency standards. On the other hand, the ETS modeled by us cannot precisely capture the current design feature of the ETS in China. At present, there is high uncertainty regarding the allowance allocation method for the future. The current design can work in the short term for the power sector which only covers one type of product, electricity, but it would have to be modified if additional sectors that produce different products are added to the system. We highlight this limitation and make it more transparent on page 13.

REVIEWERS' COMMENTS:

Reviewer #2 (Remarks to the Author):

This paper poses a very interesting question for which it is not easy to provide a quantitative answer – Is China on a path to meet its nationally determined contribution under that Paris Agreement? It addresses the question through two approaches: an expert elicitation and a systems dynamics model.

With a reported 2000 variables and a laundry list of policies to impact CO2 emissions, plus minimal discussion of the model or references about it, this paper is probably something that communicates with an inside crowd but does not communicate well with outsiders. As someone who is not familiar with systems dynamic models the numbers in the paper, sometimes to 4 significant figures, seem like something from the Ouija board.

The conclusion is presented as “conditional” on circumstances so complex that the conclusion seems in fact, to me, to be quite doubtful . There are some very big “if statements”.

The paper runs into problems on time sequence, with statements such as “slowly phase in and be fully implemented by 2020”, That is less than a year from now.

Figure captions are still inadequate, Figure 5 is inadequately referenced, units are not defined in Figure 7, Table 2 has undefined acronyms like Fit and FYP.

There is no follow-up on the paragraph 5 reference to the quality of the Chinese data.

The bottom line is that it is an interesting paper (with yet some mechanical problems) but I do not find the conclusions qualitatively or quantitatively convincing.